# Novel methods for epistasis detection in genome-wide association studies

**Lotfi Slim** [1,2]*, **Clément Chatelain**[2], **Chloé-Agathe Azencott**[1,3], **Jean-Philippe Vert** [1,4]

**1** CBIO—Centre for Computational Biology, Mines ParisTech, Paris, France, **2** Translational Sciences, SANOFI R&D, Chilly-Mazarin, France, **3** Institut Curie, PSL Research University, INSERM, U900, Paris, France, **4** Google Brain, Paris, France

* lotfi.slim@mines-paristech.fr

## Abstract

More and more genome-wide association studies are being designed to uncover the full genetic basis of common diseases. Nonetheless, the resulting loci are often insufficient to fully recover the observed heritability. Epistasis, or gene-gene interaction, is one of many hypotheses put forward to explain this missing heritability. In the present work, we propose epiGWAS, a new approach for epistasis detection that identifies interactions between a target SNP and the rest of the genome. This contrasts with the classical strategy of epistasis detection through exhaustive pairwise SNP testing. We draw inspiration from causal inference in randomized clinical trials, which allows us to take into account linkage disequilibrium. EpiGWAS encompasses several methods, which we compare to state-of-the-art techniques for epistasis detection on simulated and real data. The promising results demonstrate empirically the benefits of EpiGWAS to identify pairwise interactions.

## Introduction

Decrease in sequencing cost has widened the scope of genome-wide association studies (GWAS). Large cohorts are now built for an ever growing number of diseases. In common ones, the disease risk depends on a large number of genes connected through complex interaction networks. The classical approach and still widespread methodology in GWAS is to implement univariate association tests between each single nucleotide polymorphism (SNP) and the phenotype of interest. Such an approach is limited for common diseases, where the interactions between distant genes, or epistasis, need to be taken into account. For instance, several epistatic mechanisms have been highlighted in the onset of Alzheimer's disease [1]. Most notably, the interaction between the two genes BACE1 and APOE4 was found to be significant on four distinct datasets. Moreover, at least two epistatic interactions were also reported for multiple sclerosis [2, 3].

Several strategies [4, 5] have been developed for the detection of statistical epistasis. Many of them consist in exhaustive SNP-SNP interaction testing, followed by corrections for multiple hypothesis testing using procedures such as Bonferroni correction [6] or the Benjamini-Hochberg [7] (BH) procedure. For all procedures, the correction comes at the cost of poor statistical power [8]. For high-order interactions, the loss in statistical power is aggravated by the

**Data Availability Statement:** All relevant data are within the manuscript and its Supporting information files.

**Funding:** This study makes use of data generated by the Wellcome Trust Case-Control Consortium. A

full list of the investigators who contributed to the generation of the data is available from http://www.wtccc.org.uk. Funding for the project was provided by the Wellcome Trust under award 076113, 085475 and 090355. In the form of salaries, L. S. and C. C. were supported in their research by SANOFI and JP. V. by Google. These companies did not have any additional roles in the study design, data collection and analysis, decision to publish, or preparation of the manuscript. The specific roles of these authors are articulated in the 'author contributions' section.

**Competing interests:** The commercial affiliations of some of the co-authors do not alter our full adherence to all PLOS ONE policies on sharing data and materials.

large number of SNP tuples to consider. Moreover, exhaustive testing for high-order interactions is also accompanied by an increase in computational complexity. For increased speed, the current state-of-the-art BOOST [9] and its GPU-derivative [10] add a preliminary screening to filter non-significant interactions. Another fast interaction search algorithm in the high-dimensional setting is the xyz-algorithm [11].

By contrast, instead of constructing exhaustive models, we propose to focus on the interactions that involve a given variant, that we refer to as the target in what follows. The target is a formerly identified SNP that can be extracted from top hits in previous GWAS, causal genes, or experiments. The main rationale behind this approach is to leverage the established dependency between the target and the phenotype for a better detection of epistatic phenomena: a lower number of interactions has to be studied with the additional guarantee that the target affects the phenotype in question. In addition, focusing on interactions with a single variant allows us to model the interaction of this variant with all other SNPs in the genome at once, rather than pair of SNPs by pair of SNPs.

For the purpose of epistasis detection, the pure synergistic effects of the target with other variants must be decoupled from the marginal effects of the target and the other variants. A failure to address this issue can alter the results. One way to do so is to use an $\ell_1$-penalized regression model [12] with both marginal effect and quadratic interaction terms. If only one target SNP is investigated, generating as many quadratic interaction terms as remaining SNPs in the genome, the number of coefficients in this regression is doubled compared to a linear model with only marginal effects, rather than squared if all pairwise interaction terms were to be considered. However, this is still too many in a high-dimensional context such as GWAS. To improve the inference of the interaction coefficients, Bien et al. [13] introduced hierNET, a LASSO with hierarchy constraints between marginal and interactions terms. However, this approach does not scale to more than a hundred variables and is therefore inapplicable to GWAS data.

We turn instead towards methods developed in the context of randomized controlled trials, which aim at detecting synergies between a treatment (rather than a target SNP) and a set of covariates (rather than other SNPs) towards an outcome (rather than a phenotype). We draw on this analogy to propose two families of methods for epistasis detection. First, modified outcome approaches are inspired by the work of Tian et al. [14]. Here we construct a modified phenotype from the phenotype and all SNPs, in such a way that the SNPs in epistasis with the target form the support of a sparse linear regression between this modified phenotype and the non-target SNPs. Second, outcome weighted learning approaches are inspired by the work of Zhao et al. [15]. Here the SNPs in epistasis with the target form the support of a weighted sparse linear regression between the phenotype and the non-target SNPs, with samples weighted according to the phenotype and the target SNP.

A major difference between our setting and that of randomized controlled trials is the fact that, where they assume that the treatment is independent from the covariates, we cannot assume independence between the target SNP and the rest of the genome. Indeed, although recombination can be expected to break down non-random associations between alleles at several loci, such associations exist, and are referred to as linkage disequilibrium [16]. To account for this dependence, we introduce the equivalent of propensity scores (that is to say, the probability of treatment given the covariates [17]) in the modified outcome and outcome weighted learning approaches. In addition, the high dimensionality of the data leads us to use stability selection [18, 19] to select the regularization parameter of the $\ell_1$-penalized regressions.

In summary, we develop a new framework to study epistasis by solely focusing on the synergies with a predetermined target. By proceeding this way, our methods improve the recovery of interacting SNPs compared to standard methods like GBOOST or a LASSO with interaction terms. We demonstrate the performance of our methods against both of them for several types

of disease models. We also conduct a case study on a real GWAS dataset of type II diabetes to demonstrate the scalability of our methods and to investigate the resulting differences between them.

## Materials and methods

### Setting and notations

We jointly model genotypes and phenotypes as a triplet of random variables $(X, A, Y)$ with distribution $P$, where $Y$ is a discrete (e.g. in case-control studies) or continuous phenotype, $X = (X^{(1)}, \cdots, X^{(p)}) \in \{0, 1, 2\}^p$ represents a genotype with $p$ SNPs, and $A$ is the $(p + 1)$-th target SNP of interest. The reason why we split the $p + 1$ SNPs into $X$ and $A$ is that our goal is to detect interactions involving $A$ and other SNPs in $X$. Several selection strategies are possible for the anchor target $A$: eQTL SNPs for genes with proven effect on the phenotype $Y$, deleterious splicing variants, or among significant SNPs in previous GWAS. In classical GWAS, the SNPs are identified on the basis of the significance of their main effects. A SNP with interaction effects only can then be overlooked. To detect such SNPs, we can use association measures such as distance correlation [20] and mutual information [21] which can better capture second-order interaction effects. Alternatively, for the genotype $X$, we can choose the rest of the genome (the whole genome except the target $A$) or a given set of SNPs. The SNP set may correspond to a genomic region of interest e.g. gene, promoter region, or a pathway.

We restrict ourselves to a binary encoding of $A$ in $\{-1, +1\}$, which allows us to study both recessive and dominant phenotypes, depending on how we binarize the SNP represented in $A$. For instance, to model dominant effects, we respectively map $\{0\}$ and $\{1, 2\}$ to $\{-1\}$ and $\{+1\}$. We also introduce a second binarized version of the target SNP $A$ taking values in $\{0, 1\}$ by letting $\tilde{A} = (A + 1)/2$. SNP binarization is a common procedure in GWAS in particular for the study of epistasis. Prabhu and Pe'er [22] and Llinares-López [23] implement binarized genotypes, while Achlioptas et al. [24] use locality-sensitive hashing (LSH) to transform the original genotypes into binary vectors. The question is moot in doubled haploid organisms, where the SNPs are homozygous only.

The target SNP $A$ being symmetric and binary, it is always possible to decompose the genotype and phenotype relationship as:

$$Y = \mu(X) + \delta(X) \cdot A + \epsilon, \tag{1}$$

where $\epsilon$ is a zero mean random variable and,

$$\begin{cases} \mu(X) & = \frac{1}{2}\left[\mathbb{E}(Y|A = +1, X) + \mathbb{E}(Y|A = -1, X)\right], \\ \delta(X) & = \frac{1}{2}\left[\mathbb{E}(Y|A = +1, X) - \mathbb{E}(Y|A = -1, X)\right]. \end{cases} \tag{2}$$

If we further decompose $\delta(X) = \delta_0 + \delta_1(X)$ with $\mathbb{E}(\delta_1(X)) = 0$, then $\delta_0$ represents the main effect of $A$, and $\delta_1(X)$ the synergistic effects between $A$ and all SNPs in $X$. In the context of genomic data, we can interpret these synergies as pure epistatic effects: the main effects are accounted for by $\mu(X)$ and $\delta_0$. Furthermore, if $\delta_1(X)$ is sparse, meaning that it only depends on a subset of elements of $X$, referred to as the support of $\delta_1(X)$, then the SNPs in this support are the ones interacting with $A$. In other words, searching for epistatic interactions between $A$ and SNPs in $X$ amounts to searching for the support of $\delta$.

A GWAS dataset is a set of $n$ triplets $(X_i, A_i, Y_i)_{i=1,\ldots,n}$, which we model as independent random variables identically distributed according to $P$. To estimate the support of $\delta(X)$ from a

GWAS dataset, we propose several models based on sparse regression. The common thread between them is the use of propensity scores to estimate $\delta(X)$ and its support without estimating $\mu(X)$. We borrow the notion of propensity score from the causal inference literature, where we are interested in estimating the effect of a treatment on individuals characterized by covariates $x$. In that context, the propensity score $e(x)$ is defined as the conditional probability of being treated for an individual with covariates $x$. The propensity score can be used to compensate the differences in covariates between the two groups in observational studies, where, by contrast with randomized controlled trials, investigators have no control over the treatment assignment [25]. In our case, by analogy, we define the propensity score $e(x)$ for a configuration of SNPs $X = x$ as the probability that the target SNP $A$ is equal to 1, i.e., $e(x) = P(A = 1 | X = x)$. This score allows us to model linkage disequilibrium (LD) between $A$ and other nearby SNPs within $X$. Based on this notion of propensity score, the first family of methods we propose falls under the modified outcome banner [14]. In these models, an outcome that combines the phenotype $Y$ with the target SNP $A$ and the propensity score $e(X)$ is fit linearly to the genomic covariates $X$. We propose several variants of this approach, which differ in their control of estimation errors. Our second proposal is a case-only method based on the framework of outcome weighted learning [15]. In this model, which is a weighted binary classification problem, the outcome is the target SNP $A$, the covariates are the rest of the genotype $X$, while the phenotype $Y$ and the propensity score $e(X)$ are incorporated in the sample weights.

If not stated otherwise, the full data pipeline is written in the **R** language. The methods presented in this work are implemented in the **R** package **epiGWAS**, which is directly available via CRAN. The source code can also be downloaded from the GitHub repository https://github.com/EpiSlim/epiGWAS.

## Modified outcome regression

Depending on the underlying target value and the binarization rule, only one of the two possibilities $A = +1$ or $A = -1$ is observed for a given sample. In other words, as in randomized controlled trials where, for each sample, either the treatment is applied or it is not, here, for any given sample, we do not observe the phenotype associated with the same genotype except in A which takes the other value. Hence $\delta(X)$ cannot be estimated directly from GWAS data using Eq (2). The propensity score comes into play to circumvent this problem. By considering the new binarized variable $\tilde{A} = (A + 1)/2 \in \{0, 1\}$, we can indeed use the fact that

$$\begin{cases} \mathbb{E}[Y\tilde{A} \,|\, X] = \mathbb{E}[Y \,|\, X, \tilde{A} = 1]e(X)\,, \\ \mathbb{E}[Y(1 - \tilde{A}) \,|\, X] = \mathbb{E}[Y \,|\, X, \tilde{A} = 0](1 - e(X))\,, \end{cases}$$

to rewrite Eq (2) as:

$$\delta(X) = \frac{1}{2}\mathbb{E}\big[\tilde{Y}|X\big]\,, \tag{3}$$

where we define the modified outcome $\tilde{Y}$ of an observation $(X, A, Y)$ as:

$$\tilde{Y} = Y\left(\frac{\tilde{A}}{e(X)} - \frac{1 - \tilde{A}}{1 - e(X)}\right). \tag{4}$$

Our definition of modified outcome in Eq (4) generalizes that of Tian et al. [14], where it is defined as $\tilde{Y} = Y\tilde{A}$; both definitions are equivalent in the specific situation considered by Tian et al. [14] where $A$ and $X$ are independent, i.e., $e(x) = 1/2$ for all $x$. Our definition (4) remains

valid even when $A$ and $X$ are not independent, and can therefore accommodate the diversity of the LD landscape and of the broad range of minor allele frequencies.

Given Eq (3), we propose to estimate the support of $\delta$ from GWAS data by first transforming them into genotype-modified outcome pairs $(X_i, \tilde{Y}_i)_{i=1,...,n}$, and then applying a sparse least-squares regression model for support recovery. For that purpose, we use an elastic net linear regression model, combined with a stability selection procedure for support selection, as subsequently discussed in this paper.

In practice, however, creating the modified outcome $\tilde{Y}_i$ from a triplet $(X_i, A_i, Y_i)$ using (4) raises two issues: (i) the propensity score $e(X_i)$ must be known, and (ii) when the propensity score is close to 0 or 1, then the propensity score weighting may create numerical instability and large variance in the estimation of $\delta$. Similar problems arise in the causal inference literature, particularly for techniques based on inverse propensity score weighting techniques (IPW) [25] and we consider four standard approaches to form modified outcomes with inverse propensity score weights. They all start with an estimate $\hat{e}(X)$ of the true propensity score, which we later explain.

- Modified outcomes are simply obtained by replacing $e(X_i)$ by its estimate $\hat{e}(X_i)$ in (4):

$$\tilde{Y}_i = Y_i \left( \frac{\tilde{A}_i}{\hat{e}(X_i)} - \frac{1 - \tilde{A}_i}{1 - \hat{e}(X_i)} \right).$$

- Shifted modified outcomes are obtained by simply adding a small term $\xi = 0.1$ to the denominators in order to limit the inverse propensity score weight of each individual to a maximum of $1/\xi$:

$$\tilde{Y}_i = Y_i \left( \frac{\tilde{A}_i}{\hat{e}(X_i) + \xi} - \frac{1 - \tilde{A}_i}{1 - \hat{e}(X_i) + \xi} \right).$$

- Normalized modified outcomes are obtained by scaling differently the inverse propensity scores of individuals with $\tilde{A} = 0$ and $\tilde{A} = 1$, so that the total weights of individuals in each group is the same. This normalization was shown to be beneficial empirically for the estimation of average treatment effect in causal inference with IPW estimators [26]:

$$\tilde{Y}_i = Y_i \left( w_1 \frac{\tilde{A}_i}{\hat{e}(X_i)} - w_0 \frac{1 - \tilde{A}_i}{1 - \hat{e}(X_i)} \right),$$

where, for $t = 0, 1$,

$$w_t = \left( \sum_{j=1}^{n} t \frac{\tilde{A}_j}{\hat{e}(X_j)} + (1 - t) \frac{1 - \tilde{A}_j}{1 - \hat{e}(X_j)} \right)^{-1}.$$

- Robust modified outcomes are also borrowed from the causal inference literature, and were shown to have small large-sample variance when used for average treatment effect prediction with IPW estimators [26]:

$$\tilde{Y}_i = Y_i \left( w_1 \left( 1 - \frac{C_1}{\hat{e}(X_i)} \right) \frac{\tilde{A}_i}{\hat{e}(X_i)} - w_0 \left( 1 - \frac{C_0}{1 - \hat{e}(X_i)} \right) \frac{1 - \tilde{A}_i}{1 - \hat{e}(X_i)} \right),$$

where, for $t = 0, 1$,

$$C_t = \frac{\sum_{j=1}^{n} \frac{\tilde{A}_j - \hat{e}(X_i)}{t\hat{e}(X_i) + (t-1)(1-\hat{e}(X_i))}}{\sum_{j=1}^{n} \left[ \frac{\tilde{A}_j - \hat{e}(X_i)}{t\hat{e}(X_i) + (1-t)(1-\hat{e}(X_i))} \right]^2},$$

and

$$w_t = \left( \sum_{j=1}^{n} t \left( 1 - \frac{C_1}{\hat{e}(X_i)} \right) \frac{\tilde{A}_j}{\hat{e}(X_j)} + (1-t) \left( 1 - \frac{C_0}{1 - \hat{e}(X_i)} \right) \frac{1 - \tilde{A}_j}{1 - \hat{e}(X_j)} \right)^{-1}.$$

## Outcome weighted learning

Inspired by the outcome weighted learning (OWL) model of Zhao et al. [15], developed in the context of randomized clinical trials, we now propose an alternative to the modified outcome approach to estimate $\delta(X)$ and its support using a weighted binary classification formulation. As with OWL, this formulation mathematically amounts to predicting $A$ from $X$, where prediction errors are weighted according to $Y$ in the fitting process. In the original OWL proposal, the goal is to determine an optimal individual treatment rule $d^*$ that predicts treatment $A$ from prognostic variables $X$ so as to maximize the clinical outcome $Y$. In our context, this translates to determining an optimal predictor $d^*$ that predicts target SNP $A$ from genotype $X$, so as to maximize $Y$ (which is larger for cases than controls). We expect such a predictor to rely on the SNPs that interact with $A$ towards predicting the phenotype $Y$. We assume in this section that $Y$ only takes nonnegative values, e.g., $Y \in \{0, 1\}$ for a case-control study. To take into account the dependency between $A$ and $X$, we replace $P(A)$ with $P(A|X)$ in the original OWL definition [15] and look for the following decision rule:

$$d^* \in \underset{d:\{0,1,2\}^p \to \mathbb{R}}{\operatorname{argmin}} \mathbb{E} \left[ \frac{Y}{P(A|X)} \phi(Ad(X)) \right], \tag{5}$$

where $\phi$ is a non-increasing loss function such as the logistic loss:

$$\forall u \in \mathbb{R}, \quad \phi(u) = \log(1 + e^{-u}). \tag{6}$$

The reason to consider this formulation is that:

**Lemma 1.** *The solution $d^*$ to* (5), (6) *is:*

$$\forall x \in \{0, 1, 2\}^p, \quad d^*(x) = \ln \frac{\mathbb{E}[Y|A = +1, X = x]}{\mathbb{E}[Y|A = -1, X = x]}.$$

*Proof.* For any $x \in \{0, 1, 2\}^p$, we see from Eq (5) that $d^*(x)$ must minimize the function $l : \mathbb{R} \to \mathbb{R}$ defined by

$$\forall u \in \mathbb{R}, \quad l(u) = \mathbb{E} \left[ \frac{Y}{P(A|X = x)} \phi(Au) \,|\, X = x \right]$$

$$= \phi(u)\mathbb{E}[Y|A = 1, X = x] + \phi(-u)\mathbb{E}[Y|A = -1, X = x].$$

This function is minimized when $l'(u) = 0$, that is, when $\phi'(u)\mathbb{E}[Y|A = 1, X = x] = \phi'(-u)\mathbb{E}[Y|A = -1, X = x]$, which is equivalent to:

$$\frac{\mathbb{E}[Y|A = 1, X = x]}{\mathbb{E}[Y|A = -1, X = x]} = e^u.$$

Lemma 1 clarifies how $d^*$ is related to $\delta$ as defined in Eq (2): while $\delta$ is half the difference between the expected phenotype conditioned on the two alternative values of $A$, $d^*$ is the log-ratio of the same two quantities. In particular, both functions have the same sign for any genotype $X$. Hence we propose to estimate $d^*$ and its support, as an approximation and alternative to estimating $\delta$ and its support, in order to capture SNPs in epistasis with $A$.

For any given $(X, A, Y)$, if we define the weight $W = Y/P(A|X)$, we can interpret $d^*$ in Eq (5) as a logistic regression classifier that predicts $A$ from $X$, with errors weighted by $W$. Hence $d^*$ and its support can be estimated from GWAS data by standard tools for weighted logistic regression and support estimation. We use an elastic net logistic regression model, combined with a stability selection procedure for model selection, detailed afterwards.

In the case of qualitative GWAS studies, we encode $Y$ as 0 for controls and 1 for cases. The sample weights $W$ of controls thus become 0, resulting in a case-only approach for epistasis detection. Tools such as PLINK [27] and INTERSNP [28] similarly implement case-only analyses, which can be more powerful in practice than a joint case-control analysis [4, 29–31]. In the case of PLINK and INTERSNP, additional hypotheses such as the independence of SNP–SNP frequencies are nonetheless needed to ensure the validity of the statistical test. In our case, the family of weights $\{W_i = 1/P(A_i|X_i)\}_{i=1,\cdots,n}$ accounts for the dependency between the target $A$ and the genotype $X$. We can therefore forego such hypotheses on the data. We may even argue that the controls are indirectly included in the regression model through $P(A|X)$. It represents the dependency pattern within the general population, which consists of both cases and controls.

### Estimate of the propensity score

In causal inference, the estimation of propensity scores $e(X) = P(A = 1|X)$ is often achieved thanks to parametric models such as a logistic regression between $A$ and $X$. Because of the risk of overfitting in such an ultra high-dimensional setting, we turn instead towards hidden Markov models, which are commonly used in genetics to model linkage disequilibrium and were initially developed for imputation [32]. In this model (see the S1 File), the hidden states represent contiguous clusters of phased haplotypes. The emission states correspond to SNPs.

Since the structural dependence is chromosome-wise, we only retain the SNPs located on the same chromosome as the SNP $A$—which we denote here by $X_A$—for the estimate of $P(A|X)$. Mathematically, this is equivalent to the independence of the SNPs $A$ and $X_A$ from the SNPs of other chromosomes.

The pathological cases $P(A|X_A) \approx 1$ and $P(A|X_A) \approx 0$ can be avoided by the removal of all SNPs within a certain distance of $A$. In our implementation, we first perform an adjacency-constrained hierarchical clustering of the SNPs located on the chromosome of the target $A$. We fix the maximum correlation threshold at 0.5. To alleviate strong linkage disequilibrium, we then discard all neighboring SNPs within a three-cluster window of SNP $A$. Such filtering is sensible since we are looking for biological interactions between functionally-distinct regions. The neighboring SNPs are not only removed for the estimation of the propensity score, but also in the regression models searching for interactions.

After the filtering and the fitting of the unphased genotype model using fastPHASE, the last remaining step is the application of the forward algorithm [33] to obtain an estimate of the two potential observations $(A = 1, X_A)$ and $(A = -1, X_A)$. Bayes theorem then yields the desired probability $P(A|X) = P(A|X_A) = P(A, X_A)/(P(A = +1, X_A) + P(A = -1, X_A))$.

### Support estimation

In order to estimate the support of $\delta$ in the case of modified outcome regression (3), and of $d^*$ in the case of OWL (5), we model both functions as linear models and estimate

non-zero coefficients by elastic net regression [34] combined with stability selection [18, 19].

More precisely, given a GWAS cohort $(X_i, A_i, Y_i)_{i=1,\cdots,n}$, we first define empirical risks for a candidate linear model $x \mapsto \gamma^\top x$ for $\delta$ and $d^*$ as respectively

$$R_1(\gamma) = \frac{1}{n}\sum_{i=1}^{n}\left(\tilde{Y}_i - \gamma^\top X_i\right)^2, \quad R_2(\gamma) = \frac{1}{n}\sum_{i=1}^{n}\frac{Y_i}{P(A_i|X_i)}\phi(A_i\gamma^\top X_i).$$

For a given regularization parameter $\lambda > 0$ and empirical risk $R = R_1$ or $R = R_2$, we then define the elastic net estimator:

$$\hat{\gamma}_\lambda \in \underset{\gamma}{\operatorname{argmin}} \quad R(\gamma) + \lambda\left[(1-s)\|\gamma\|_1 + \frac{1}{2}s\|\gamma\|_2^2\right],$$

where we fix $s = 10^{-6}$ to give greater importance to the $L_1$-penalization. Over a grid of values $\Lambda$ for the penalization parameter $\lambda$, we subsample $N = 50$ times without replacement over the whole cohort. The size of the generated subsamples $I_1, \cdots, I_N$ is $\lfloor n/2 \rfloor$. Each subsample $I$ provides a different support for $\hat{\gamma}_\lambda$, which we denote $\hat{S}^\lambda(I)$. For $\lambda \in \Lambda$, the empirical frequency of the variable $X_k$ entering the support is then given by:

$$\hat{\omega}_k^\lambda = \frac{1}{N}\sum_{j=1}^{N}1(k \in \hat{S}^\lambda(I_j)).$$

In the original stability selection procedure [18], the decision rule for including the variable $k$ in the final model is $\max_{\lambda \in \Lambda}\hat{\omega}_k^\lambda \geq t$. The parameter $t$ is a predefined threshold. For noisy high-dimensional data, the maximal empirical frequency along the stability path $\max_{\lambda \in \Lambda}\hat{\omega}_k^\lambda$ may not be sufficiently robust because of its reliance on a single noisy measure of $\hat{\omega}_k^\lambda$ to derive the maximum. Instead, we used the area under the stability path, $\int_\lambda \hat{\omega}_k^\lambda \, d\lambda$, as propsed by Haury et al. [19]. The main intuition behind the better performance is the early entry of causal variables into the LASSO path.

Finally, to determine the grid $\Lambda$, we use the **R** package **glmnet** [35]. We generate a log-scaled grid of 200 values $(\lambda_l)_{l=1,\cdots,200}$ between $\lambda_1 = \lambda_{max}$ and $\lambda_{200} = \lambda_{max}/100$, where $\lambda_{max}$ is the maximum $\lambda$ leading to a non-zero model. To improve inference, we only retain the first half of the path comprised between $\lambda_1$ and $\lambda_{100}$. The benefit of a thresholded regularization path is to discard a large number of irrelevant covariates that enter the support for low values of $\lambda$.

## Results

### Simulations

**Disease model.** We simulate phenotypes using a logit model with the following structure:

$$\operatorname{logit}(P(Y = 1|\tilde{A} = i, X)) = \beta_{i,V}^T X_V + \beta_W^T X_W + X_{Z_1}^T \operatorname{diag}\left(\beta_{Z_1,Z_2}\right)X_{Z_2},$$

where $V$, $W$, $Z_1$ and $Z_2$ are random subsets of $\{1, \cdots, p\}$. The variables within the vector $X_V$ interact with $A$. The variables in $X_W$ corresponds to marginal effects, while $X_{Z_1}$ and $X_{Z_2}$ correspond to pairs of quadratic effects between SNPs that exclude $A$. The effect sizes $\beta_{0,V}, \beta_{1,V}, \beta_W$ and $\beta_{Z1,Z_2}$ are sampled from $\mathcal{N}(0, 1)$. Given the symmetry around 0 of the effect size distributions, the simulated cohorts are approximately equally balanced between cases and controls.

To account for the diversity of effect types in disease models, we simulate four scenarios with different overlap configurations between $X_V$ and $(X_W, X_{Z_1})$:

- Synergistic only effects, $|V \cap W| = 0$, $|V \cap Z_1| = 0$, $|V| = |W| = |Z_1| = |Z_2| = 8$;

- Partial overlap between synergistic and marginal effects, $|V \cap W| = 4$, $|V \cap Z_1| = 0$, $|V| = |W| = |Z_1| = |Z_2| = 8$;

- Partial overlap between synergistic and quadratic effects, $|V \cap W| = 0$, $|V \cap Z_1| = 4$, $|V| = |W| = |Z_1| = |Z_2| = 8$;

- Partial overlap between synergistic and quadratic/marginal effects, $|V \cap W| = 2$, $|V \cap Z_1| = 2$, $|V| = |W| = |Z_1| = |Z_2| = 8$.

For each of the above scenarios, we conduct 125 simulations: 5 sets of causal SNPs $\{A, V, W, Z_1, Z_2\} \times 5$ sets of size effects $\{\beta_{0,V}, \beta_{1,V}, \beta_W, \beta_{Z1,Z2}\} \times 5$ replicates. Within each scenario, we consider multiple SNP sets to model the range of MAFs and LD which can exist between $A$ and $X$.

Because of the filtering window around the SNP $A$, the causal SNPs $(X_V, X_W, Z_1, Z_2)$ are sampled outside of that window. The second constraint on the causal SNPs is a lower bound on the minor allele frequencies (MAF). We fix that bound at 0.2. The goal is to obtain well-balanced marginal distributions for the different variants. For rare variants, it is difficult to untangle the statistical power of any method from the inherent difficulty in detecting them. The lower bound is also coherent with the common disease-common variant hypothesis [36]: the main drivers of complex/common diseases are common SNPs.

**Genotype simulations.** For the sake of coherence, we simulate genotypes using the second release of HAPGEN [37]. The underlying model for HAPGEN is the same hidden Markov model used in fastPHASE. The starting point of the simulations is a reference set of population haplotypes. The accompanying haplotypes dataset is the 1000 Genomes phase 3 reference haplotypes [38]. In our simulations, we only use the European population samples. The second input to HAPGEN is a fine scale recombination map. Consequently, the simulated haplotypes/genotypes exhibit the same linkage disequilibrium structure as the original reference data.

In comparison to the HAPGEN-generated haplotypes, the markers density for SNP arrays is significantly lower. For example, the sequencing technology for the WTCCC case-control consortium [39] is the Affymetrix 500K. As its name suggests, "only" five hundred thousand positions are genotyped. As most GWAS are based on SNP array data, we only extract from the simulated genotypes the markers of the Affymetrix 500K. In the subsequent QC step, we only retain common bi-allelic SNPs defined by a MAF $> 0.01$. We also remove SNPs that are not in a Hardy-Weinberg equilibrium ($p < 10^{-6}$). We do not conduct any additional LD pruning for the SNPs in $X$. For univariate GWAS, LD pruning reduces dimensionality while approximately maintaining the same association patterns between genotype and phenotype. For second order interaction effects, the loss of information can be more dramatic, as the retained SNP pairs can be insufficient to represent the complex association of corresponding genomic regions with the phenotype.

For iterative simulations, HAPGEN can be time-consuming, notably for large cohorts consisting of thousands of samples. We instead proceed in the following way: we generate once and for all a large dataset of 20 thousand samples on chromosome 22. To benchmark for varying sample sizes $n \in \{500, 1000, 2000, 5000\}$, we iteratively sample uniformly and without replacement $n$-times the population of 20000 individuals to create 125 case-control cohorts. On chromosome 22, we then select $p = 5000$ SNPs located between the nucleotide positions 16061016 and 49449618. We do not conduct any posterior pruning to avoid filtering out the true causal SNPs.

**Evaluation.** We benchmark our new methods against two baselines. The first method is GBOOST [9], a state-of-the-art method for epistasis detection. For each SNP pair, it

implements the log-likelihood ratio statistic to compare the goodness of fit of two models: the full logistic regression model with both main effect and interaction terms, and the logistic regression model with main effects only. The preliminary sure screening step in GBOOST to discard a number of SNPs from exhaustive pairwise testing was omitted, since we are only interested in the ratio statistic for all pairs of the form $(A, X_k)$, where $X_k$ is the $k$-th SNP in $X$. The second method, which we refer to as product LASSO, originates from the machine learning community. It was developed by Tian et al. [14] to estimate interactions between a treatment and a large number of covariates. It fits an $L_1$-penalized logistic regression model with $A \times X$ as covariates. The variable of interest $A$ is symmetrically encoded as $\{-1, +1\}$. Under general assumptions, Tian et al. [14] show how this model works as a good approximation to the optimal decision rule $d^*$.

We visualize the support estimation performance in terms of receiver-operating characteristic (ROC) curves and precision-recall (PR) curves. For a particular method in a given scenario, a single ROC (resp. PR) curve allows to visualize the ability of the algorithm to recover causal SNPs. For each SNP, the prediction score is the area under its corresponding stability path. The ground truth label is 1 for the SNPs interacting with the target $A$, and 0 otherwise. In the high-dimensional setting of GWAS, the use of raw scores instead of $p$-values lends more robustness to our methods, by avoiding finite-sample approximations of the score distributions and multiple hypothesis corrections.

The covariates and the outcome differ between our methods. That implies a different regularization path for each method and as a result, incomparable stability paths. For better interpretability and comparability between the methods, we use the position $l$ on the stability path grid $\Lambda = (\lambda_l)$ s.t. $\lambda_l > \lambda_{l+1}$ instead of the value of $\lambda_l$ for computing the area under the curve.

In Fig 1, we provide the ROC and PR curves for the fourth scenario which corresponds to a partial overlap between synergistic and quadratic/marginal effects and for a sample size $n = 500$. Because of space constraints, all ROC/PR figures and corresponding AUC tables are listed in the S2 File. The figures represent the average ROC and PR curves of the 125 simulations in each of the four scenarios. To generate those figures, we used the **R** package **precrec** [40]. It performs nonlinear interpolation in the PR space. The AUCs are computed with same package.

Regardless of the scenario and the sample size, the areas under all ROC curves are higher than 0.5. This confirms that all of them perform better than random, yet with varying degrees of success. By contrast, the overall areas under the precision-recall curves are low. The maximum area under the precision-recall curve is 0.41, attained by modified outcome with shifted weights for $n = p$. This can be attributed to the imbalanced nature of the problem: 8 synergistic SNPs out of 5000. We also check that the AUCs increase with the cohort size for both ROC and PR domains.

The best performing methods are robust modified outcome and GBOOST. Robust modified outcome has a slight lead in terms of ROC AUCs, notably for low sample sizes. The latter setup is the closest to our intended application in genome-wide association studies. Of special interest to us in the ROC space is the bottom-left area. It reflects the performance of highly-ranked instances. For all scenarios, we witness a better start for robust modified outcome. The other methods within the modified outcome family behave similarly. Such a result was expected because of their theoretical similarities. Despite the model misspecification, product LASSO performs rather well. On average, it comes third to GBOOST and robust modified outcome. The outcome weighted learning approach which is an approximation to estimating the sign of $\delta$ has consistently been the worst performer in the ROC space.

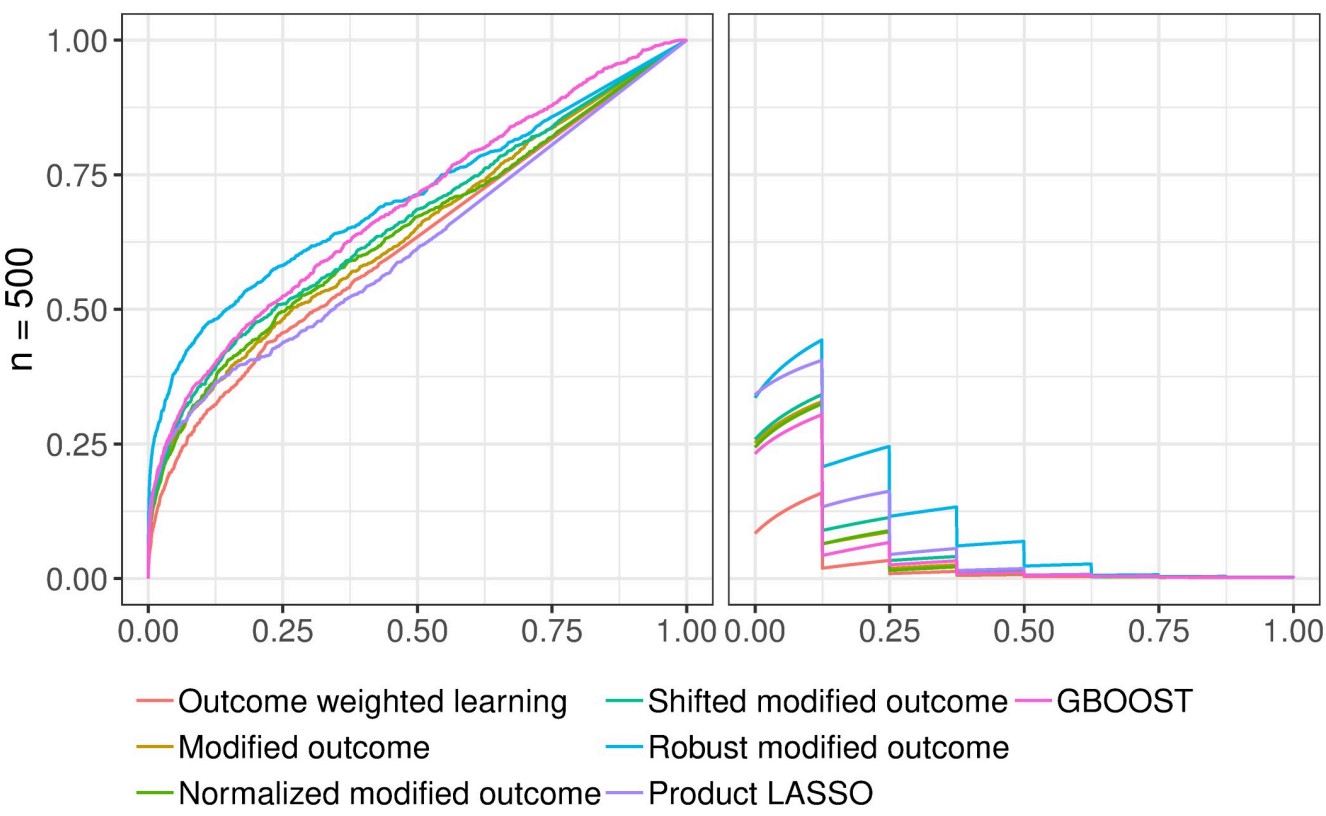

**Fig 1. Average ROC (left) and PR (right) curves for the fourth scenario and *n* = 500.**

In PR space, the results are more mixed. For low sample sizes, robust modified outcome is still the best performing method. As the sample size increases, we observe that other methods within the modified outcome family, notably shifted modified outcome, surpass the robust modified outcome approach. Surprisingly, the good performance of GBOOST in ROC space was not reproduced in PR space. This might be explained by the highly imbalanced nature of the problem and the lower performance of GBOOST, compared to robust modified outcome in the high specificity region of the ROC curves (lower left). By contrast, product LASSO is always trailing the best performer of the modified outcome family. As for ROC curves, we are also interested in the beginning of the PR curves. For a recall rate of 0.125, the highest precision rate is near 0.5 for the first, third and fourth scenarios. That implies that we detect on average one causal SNP in the first two SNPs. For the second scenario, the highest precision rate is even higher at approximately 0.68. The area under the stability path is then a robust score for model selection in the high dimensional setting.

It is worth noting the homogeneous behavior of the different methods across the four scenarios. For a given sample size, and for a given method, the ROC and PR AUCs are similar. This suggests they all successfully filtered out the common effects term $\mu(X)$ even in presence of an overlap between the causal SNPs within $\mu(X)$ and $\delta(X)$.

## Case study: Type II diabetes dataset of the WTCCC

As a case study, we selected the type II diabetes dataset of the WTCCC [39] to illustrate the scalability of our methods to real datasets. To the best of our knowledge, no confirmed

epistatic interactions exist for type II diabetes. We instead propose to study the synergies with a particular target: $rs$41475248 on chromosome 8. We focus on this target SNP because (i) GBOOST finds that it is involved in 3 epistatic interactions, when controlling for a false discovery rate of 0.05, and (ii) it is a common variant, with a MAF of 0.45.

Before running our methods on the WTCCC dataset, we applied the same QC procedures with the following thresholds: 0.01 for minor-allele frequencies and $p > 10^{-6}$ for the Hardy-Weinberg equilibrium. No additional pruning is performed. The number of remaining variants is 354439 SNPs. The number of samples is 4897, split between 1953 cases and 2944 controls.

To solve the different $L_1$-penalized regressions, we abandoned the **glmnet** package in favor of another one, **biglasso** [41]. Indeed, **glmnet** does not accept as input such ultra-high dimensional design matrices. On the other hand, **biglasso** was specifically developed for similar settings thanks to its multi-threaded implementation and utilization of memory-mapped files. Because **biglasso** does not implement sample weighting, it cannot be used to run outcome weighted learning. Since this approach performs worse than the modified outcome approaches on simulated data, we simply exclude it from this case study.

The main difficulty for the evaluation of GWAS methods is the biological validation of the study results. We often lack evidence to correctly label each SNP as being involved or not in an epistatic interaction. Evaluating the real model selection performance of the different methods on real datasets is then impossible. However, we can study the concordance between them. A common way to proceed is Kendall's tau which is a measure of rank correlation. In Table 1, we give the correlation matrix of our four variants of modified outcome methods, and of the two baseline methods GBOOST and product LASSO. All elements are positive which indicates a relative agreement between the methods. While methods using different mathematical definitions of epistasis cannot be expected to return the same results, those with similar or identical underlying models should capture similar genetic architectures and return more similar results. Modified outcome, normalized modified outcome and shifted modified outcome have the highest correlation coefficients. Such a result was expected because of their theoretical similarities. We also note that the lowest score is for robust modified outcome and GBOOST. In the previous section, these two methods were the best performing. This suggests those two methods can make different true discoveries.

In any follow-up work, we will only exploit the highly-ranked variants. A weighted tau statistic that assigns a higher weight to the first instances is therefore more relevant. Weighted nonnegative tau statistics better assess the relative level of concordance between different pairs of methods, while the sign in Kendall's tau shows if two methods rather agree or disagree. In Table 2, we list Kendall's tau coefficients with multiplicative hyperbolic weighting. Similarly, we notice that robust modified outcome is least correlated with GBOOST and most correlated with product LASSO.

Aside from rank correlation, another option to appraise the results is to measure the association between the top SNPs for each method and the phenotype. Table 3 lists the Cochran-Armitage test $p$-values for the top 25 SNPs for each method in an increasing order. Despite being synthetic univariate measures, the Cochran-Armitage statistics give us an indication of the true ranking performance. Robust modified outcome is clearly the method with the lowest $p$-values. For instance, the top 14 SNPs have a $p$-value lower than 0.001. That confirms the result of our simulations that robust modified outcome is the best performer for capturing causal SNPs. The $p$-values associated to product LASSO and GBOOST are also relatively low, with respectively 5 and 4 $p$-values lower than 0.001. However, we note the overall difficulty in drawing clear conclusions for all methods. Without multiple testing correction, most of the $p$-values for each method already exceed classical significance levels *e.g.* 0.05. For 3 out of 6

**Table 1. Concordance between methods used to determine SNPs synergistic to rs41475248 in type II diabetes, measured by Kendall's tau.**

|  | GBOOST | Modified outcome | Normalized modified outcome | Shifted modified outcome | Robust modified outcome | Product LASSO |
|---|---|---|---|---|---|---|
| GBOOST | 1.000 | 0.200 | 0.203 | 0.202 | 0.070 | 0.152 |
| Modified outcome | 0.200 | 1.000 | 0.411 | 0.405 | 0.150 | 0.283 |
| Normalized modified outcome | 0.203 | 0.411 | 1.000 | 0.406 | 0.153 | 0.284 |
| Shifted modified outcome | 0.202 | 0.405 | 0.406 | 1.000 | 0.179 | 0.301 |
| Robust modified outcome | 0.070 | 0.150 | 0.153 | 0.179 | 1.000 | 0.257 |
| Product LASSO | 0.152 | 0.283 | 0.284 | 0.301 | 0.257 | 1.000 |

methods, the $p$-values of the 25[th] SNP are greater than 0.90. Nonetheless, the existence of such high $p$-values further demonstrates the capacity of our methods in discovering novel associations undetected by univariate methods.

## Discussion

In this paper, we have proposed several methods, inspired from the causal inference literature, to select SNPs having synergystic effects with a particular target SNP towards a phenotype. The consistency of our results across the four disease models show that the proposed methods are rather successful. Indeed, their performance is not strongly impacted by the presence/absence of other marginal and epistatic effects. Among the methods we propose, robust modified outcome is the most suited to real GWAS applications. Its superior performance is partially due to its robustness against propensity score misspecification. The AUCs for robust modified outcome are overall the highest in addition to its retrieval performance for highly-ranked instances. More importantly, robust modified outcome outperforms GBOOST and other regression-based methods. This is particularly true for small number of samples ($n = 500$), which is the closest setup to real GWAS datasets. However, the low PR AUCs show that there is still room for improvement. The highest observed PR AUC is 0.17. Interestingly, we note that several of our methods clearly outperform GBOOST across all scenarios and all sample sizes in the PR space. Nonetheless, GBOOST behaves similarly to our methods in the ROC space. Such differences between ROC and PR curves are common for highly-imbalanced datasets where PR curves are more informative and discriminative [42].

In our simulations, ROC and PR AUCs were relatively close between all methods. On the other hand, according to two rank correlation measures (Kendall's tau and weighted Kendall's tau), the results do not strongly overlap between the different methods (values far from 1). For

**Table 2. Concordance between methods used to determine SNPs synergistic to rs41475248 in type II diabetes, measured by Kendall's tau with multiplicative weights.**

|  | GBOOST | Modified outcome | Normalized modified outcome | Shifted modified outcome | Robust modified outcome | Product LASSO |
|---|---|---|---|---|---|---|
| GBOOST | 1.000 | 0.483 | 0.481 | 0.517 | 0.423 | 0.501 |
| Modified outcome | 0.483 | 1.000 | 0.851 | 0.857 | 0.462 | 0.586 |
| Normalized modified outcome | 0.481 | 0.851 | 1.000 | 0.860 | 0.467 | 0.594 |
| Shifted modified outcome | 0.517 | 0.857 | 0.860 | 1.000 | 0.504 | 0.603 |
| Robust modified outcome | 0.423 | 0.462 | 0.467 | 0.504 | 1.000 | 0.596 |
| Product LASSO | 0.501 | 0.586 | 0.594 | 0.603 | 0.596 | 1.000 |

**Table 3. Cochran-Armitage test *p*-values for the top 25 SNPs for each method.**

| GBOOST | Modified outcome | Normalized modified outcome | Shifted modified outcome | Robust modified outcome | Product LASSO |
|---|---|---|---|---|---|
| **0.0000047** | **0.0000000** | **0.0000000** | **0.0000000** | **0.0000000** | **0.0000047** |
| **0.0002632** | **0.0000015** | **0.0000015** | **0.0000015** | **0.0000000** | **0.0000075** |
| **0.0002667** | **0.0002667** | **0.0002667** | **0.0002667** | **0.0000001** | **0.0000172** |
| **0.0006166** | 0.0027308 | 0.0027308 | 0.0027308 | **0.0000012** | **0.0002667** |
| 0.0015069 | 0.0093734 | 0.0093734 | 0.0093734 | **0.0000049** | **0.0005286** |
| 0.0028872 | 0.0633055 | 0.0633055 | 0.0633055 | **0.0000059** | 0.0110392 |
| 0.0031533 | 0.0724198 | 0.0724198 | 0.0724198 | **0.0000075** | 0.0122543 |
| 0.0034323 | 0.0925877 | 0.0925877 | 0.0771170 | **0.0000172** | 0.0152912 |
| 0.0081128 | 0.1126164 | 0.1043632 | 0.0925877 | **0.0002030** | 0.0346055 |
| 0.0093734 | 0.1272777 | 0.1126164 | 0.1126164 | **0.0002667** | 0.0347964 |
| 0.0142695 | 0.2552284 | 0.1567974 | 0.1272777 | **0.0003047** | 0.0396448 |
| 0.0633055 | 0.2926915 | 0.2971396 | 0.1639805 | **0.0004643** | 0.0396932 |
| 0.0771170 | 0.3436741 | 0.3529366 | 0.2971396 | **0.0005286** | 0.0527104 |
| 0.1616393 | 0.3529366 | 0.5012038 | 0.3529366 | **0.0005841** | 0.0633055 |
| 0.2089538 | 0.5871432 | 0.5506690 | 0.5012038 | 0.0015214 | 0.0763114 |
| 0.2114803 | 0.5985624 | 0.5985624 | 0.5707955 | 0.0016353 | 0.1126164 |
| 0.2256368 | 0.6016953 | 0.7183847 | 0.5985624 | 0.0025709 | 0.1185275 |
| 0.2586186 | 0.6361937 | 0.7199328 | 0.7000506 | 0.0064196 | 0.1796624 |
| 0.2654530 | 0.7183847 | 0.7342897 | 0.7183847 | 0.0080405 | 0.2552284 |
| 0.4105146 | 0.7342897 | 0.7656055 | 0.7342897 | 0.0110392 | 0.3308890 |
| 0.4323674 | 0.7979653 | 0.7706524 | 0.7979653 | 0.0122543 | 0.3867409 |
| 0.4376669 | 0.8683271 | 0.7979653 | 0.7993838 | 0.0124442 | 0.5045073 |
| 0.4796214 | 0.8820292 | 0.7993838 | 0.8683271 | 0.0136452 | 0.5985624 |
| 0.5871432 | 0.9188037 | 0.8820292 | 0.8821872 | 0.0346055 | 0.6238335 |
| 0.9479547 | 0.9903334 | 0.8821872 | 0.9188037 | 0.0396932 | 0.8821872 |

instance, GBOOST least agrees with robust modified outcome. However, the two methods are the best performing in our simulations. Different approaches seem to discover different types of interactions [43]. We conclude that a consensus method combining GBOOST and robust modified outcome could better improve the recovery of interacting SNPs.

Across all simulation settings, OWL was the worst-performing method. This is expected given the fact that OWL is only a sign approximation to modified outcome. In OWL, the common effects term is not completely filtered out, which explains the observed gap in performance between the two approaches. Despite this limitation, OWL remains statistically valid, in the sense that its support recovery performance increases with the sample size (see the S2 File). In practical settings, we naturally recommend the use of the modified outcome approaches which linearity allow a complete filtering of common effects, and consequently yield a better performance.

The carried simulations prove that the highly-ranked SNPs include false positives. This is accentuated by the imbalanced nature of our problem: a handful of causal SNPs for thousands of referenced SNPs. Hopefully, the continual decrease in genotyping costs will result in a dramatic increase in sample sizes and, in consequence, statistical power. For instance, the UK Biobank [44] comprises full genome-wide data for five hundred thousand individuals.

The case study that we carried for type II diabetes demonstrates the scalability of our methods to real GWAS. To reduce runtime, one can reduce the number of subsamples used for stability selection; however this may come at the expense of performance. The development of

new and faster LASSO solvers [45, 46] for large scale problems will further help broaden the adoption of our methods by end-users without compromising statistical performance.

The main contribution of our work is extending the causal inference framework to epistasis detection by developing a new family of methods. They rely on propensity scores to detect interactions with specific SNP targets. Given our partial understanding of common diseases and the overall lack of statistical power of existing tools, such refocused models can be more useful to further our understanding of disease etiologies. Hundreds of genes have already been associated with several diseases via univariate GWAS. The next step is to leverage such findings to detect additional synergies between these genes and the rest of the genome. Beyond a better understanding of disease mechanisms through new biomarker discovery, we see the development of combination drug therapies as an additional application of our work.

A first area of future improvement for our methods is propensity score estimation, which can benefit from a large number of recent methods [47]. A second area is incorporating multiple covariates (whether clinical covariates, variables encoding population structure or other genetic variants) to account for, among other things, higher-order interactions and population structure. A straightforward solution is to include additional variables in $X$, which encode for the other covariates. However, this will impact the consistency and interpretability of the propensity scores. A second potential solution is the use of modified targets which combine the original target with the other covariates e.g. target $\times$ gender. We think that such outcomes have not been explored because of the insufficiency of the representation by a single binary variable. To address this issue we can, for example, borrow some of the ideas in VanderWeele and Hernan [48] to construct richer representations.

## Supporting information

**S1 File. Genotypic hidden Markov model.**
(PDF)

**S2 File. Simulation results.**
(PDF)

## Author Contributions

**Conceptualization:** Clément Chatelain, Chloé-Agathe Azencott, Jean-Philippe Vert.

**Data curation:** Lotfi Slim.

**Formal analysis:** Lotfi Slim, Clément Chatelain, Chloé-Agathe Azencott, Jean-Philippe Vert.

**Investigation:** Lotfi Slim, Chloé-Agathe Azencott, Jean-Philippe Vert.

**Methodology:** Lotfi Slim, Chloé-Agathe Azencott, Jean-Philippe Vert.

**Software:** Lotfi Slim.

**Supervision:** Clément Chatelain, Chloé-Agathe Azencott, Jean-Philippe Vert.

**Validation:** Clément Chatelain, Chloé-Agathe Azencott, Jean-Philippe Vert.

**Visualization:** Lotfi Slim, Chloé-Agathe Azencott, Jean-Philippe Vert.

**Writing – original draft:** Lotfi Slim.

**Writing – review & editing:** Lotfi Slim, Chloé-Agathe Azencott, Jean-Philippe Vert.

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
