## [Decision Letter · Decision Letter 0]

15 Oct 2020

PONE-D-20-16158

Novel methods for epistasis detection in genome-wide association studie

PLOS ONE

Dear Dr. Slim,

Thank you for submitting your manuscript to PLOS ONE. After careful consideration, we feel that it has merit but does not fully meet PLOS ONE’s publication criteria as it currently stands. Therefore, we invite you to submit a revised version of the manuscript that addresses the points raised during the review process.

We look forward to receiving your revised manuscript.

Kind regards,

Jiang Gui

Academic Editor

PLOS ONE

Journal Requirements:

"Funding for the project was provided by the Wellcome Trust under award 076113, 085475 and 090355."

"No specific funding"

We note that one or more of the authors are employed by a commercial company: SANOFI, Google.

3.1. Please provide an amended Funding Statement declaring this commercial affiliation, as well as a statement regarding the Role of Funders in your study. If the funding organization did not play a role in the study design, data collection and analysis, decision to publish, or preparation of the manuscript and only provided financial support in the form of authors' salaries and/or research materials, please review your statements relating to the author contributions, and ensure you have specifically and accurately indicated the role(s) that these authors had in your study. You can update author roles in the Author Contributions section of the online submission form.

3.2. Please also provide an updated Competing Interests Statement declaring this commercial affiliation along with any other relevant declarations relating to employment, consultancy, patents, products in development, or marketed products, etc.  

Reviewers' comments:

Reviewer's Responses to Questions

**Comments to the Author**

1. Is the manuscript technically sound, and do the data support the conclusions?

Reviewer #1: Yes

2. Has the statistical analysis been performed appropriately and rigorously? 

Reviewer #1: Yes

3. Have the authors made all data underlying the findings in their manuscript fully available?

Reviewer #1: Yes

4. Is the manuscript presented in an intelligible fashion and written in standard English?

Reviewer #1: Yes

5. Review Comments to the Author

Reviewer #1: This paper is structured clearly and well written. It proposed two models for the detection of interaction effect between SNPs, of which the first one is based on the propensity scores and the second one is based on an approximation to objective function ¬\\delta\\left(x\\right).

My major concern is about the second model. A prerequisite of the second model is that the function \\delta\\left(x\\right) and d^\\star share the same support. Just as the authors said in lines 184-187, on page 7/19, “ \\delta\\left(x\\right) is half the difference between the expected phenotype , while d^\\star is the log-ratio of the same two quantities”, in other words, \\delta\\left(x\\right) is linear while d^\\star is nonlinear. Consequently, the mean function \\mu\\left(x\\right) can be cancelled from \\delta\\left(x\\right) while can not from d^\\star, which will cause the support of these two function different. The author should clarify this obscurity.

6. PLOS authors have the option to publish the peer review history of their article (what does this mean?). If published, this will include your full peer review and any attached files.

Reviewer #1: **Yes: **Jie Zhou

---

## [Author Response · Author response to Decision Letter 0]

22 Oct 2020

Dear editor,

We sincerely thank you for managing the whole review process. We also thank the reviewer for examining our manuscript and providing interesting feedback.

We would like to transfer the exact statement we included in the acknowledgement section of our original manuscript to the funding section. We also apologize for not having clarified the commercial affiliations of some of the co-authors. The amended funding statement is the following:

"This study makes use of data generated by the Wellcome Trust Case-Control 509 Consortium. A full list of the investigators who contributed to the generation of the 510 data is available from http://www.wtccc.org.uk. Funding for the project was provided by the 511 Wellcome Trust under award 076113, 085475 and 090355.

In the form of salaries, L. S. and C. C. were supported in their research by SANOFI and JP. V. by Google. These companies did not have any additional roles in the study design, data collection and analysis, decision to publish, or preparation of the manuscript. The specific roles of these authors are articulated in the ‘author contributions’ section."

Additionally, as you suggested, our competing interests statement can be updated as follows:

"The commercial affiliations of some of the co-authors do not alter our full adherence to all PLOS ONE policies on sharing data and materials."

The rest of our manuscript now meets the journal's style guidelines. We also updated the author affiliations. 

The remark of the reviewer absolutely explains the gap in performance between the modified outcome (MO) and the outcome weighted learning (OWL) approaches. In OWL, the term $d^*$ has the same sign as the term $\\delta$ in MO. The sign commonality is not sufficient to filter out the common effects term $\\mu(X)$. Despite this limitation, the OWL approach remains statistically valid, in the sense that its support recovery performance increases with the sample size (see supplementary material). In practical settings, we naturally recommend the use of the MO approaches which linearity allow a complete filtering of the term $\\mu(X)$, and consequently yield a better performance. We included this remark to our discussion section. 

We remain at our disposal for any further enquiries. 

Sincerely,

Lotfi Slim on behalf of all co-authors

---

## [Decision Letter · Decision Letter 1]

12 Nov 2020

Novel methods for epistasis detection in genome-wide association studies

PONE-D-20-16158R1

Dear Dr. Slim,

We’re pleased to inform you that your manuscript has been judged scientifically suitable for publication and will be formally accepted for publication once it meets all outstanding technical requirements.

Kind regards,

Jiang Gui

Academic Editor

PLOS ONE

Additional Editor Comments (optional):

Reviewers' comments:

Reviewer's Responses to Questions

**Comments to the Author**

1. If the authors have adequately addressed your comments raised in a previous round of review and you feel that this manuscript is now acceptable for publication, you may indicate that here to bypass the “Comments to the Author” section, enter your conflict of interest statement in the “Confidential to Editor” section, and submit your "Accept" recommendation.

Reviewer #1: All comments have been addressed

2. Is the manuscript technically sound, and do the data support the conclusions?

Reviewer #1: Yes

3. Has the statistical analysis been performed appropriately and rigorously? 

Reviewer #1: Yes

4. Have the authors made all data underlying the findings in their manuscript fully available?

Reviewer #1: Yes

5. Is the manuscript presented in an intelligible fashion and written in standard English?

Reviewer #1: Yes

6. Review Comments to the Author

Reviewer #1: I think the authors have addressed my concerns and so I agree the publication of this paper on PLOS ONE.

7. PLOS authors have the option to publish the peer review history of their article (what does this mean?). If published, this will include your full peer review and any attached files.

Reviewer #1: No

---

## [Editor Report · Acceptance letter]

16 Nov 2020

PONE-D-20-16158R1 

Novel methods for epistasis detection in genome-wide association studies  

Dear Dr. Slim:

I'm pleased to inform you that your manuscript has been deemed suitable for publication in PLOS ONE. Congratulations! Your manuscript is now with our production department. 

Kind regards, 

on behalf of

Dr. Jiang Gui 

Academic Editor

PLOS ONE